# Production of a hybrid capacitive storage device via hydrogen gas and carbon electrodes coupling

Zhengxin Zhu[1,2], Zaichun Liu ⬤ [1,2], Yichen Yin[1], Yuan Yuan[1], Yahan Meng[1], Taoli Jiang[1], Qia Peng[1], Weiping Wang[1] & Wei Chen ⬤ [1✉]

Conventional electric double-layer capacitors are energy storage devices with a high specific power and extended cycle life. However, the low energy content of this class of devices acts as a stumbling block to widespread adoption in the energy storage field. To circumvent the low-energy drawback of electric double-layer capacitors, here we report the assembly and testing of a hybrid device called electrocatalytic hydrogen gas capacitor containing a hydrogen gas negative electrode and a carbon-based positive electrode. This device operates using pH-universal aqueous electrolyte solutions (i.e., from 0 to 14) in a wide temperature range (i.e., from − 70 °C to 60 °C). In particular, we report specific energy and power of 45 Wh kg$^{-1}$ and 458 W kg$^{-1}$ (both values based on the electrodes' active materials mass), respectively, at 1 A g$^{-1}$ and 25 °C with 9 M H$_3$PO$_4$ electrolyte solution. The device also enables capacitance retention of 85% (final capacitance of about 114 F g$^{-1}$) after 100,000 cycles at 10 A g$^{-1}$ and 25 °C with 1 M phosphate buffer electrolyte solution.

[1] Department of Applied Chemistry, School of Chemistry and Materials Science, Hefei National Research Center for Physical Sciences at the Microscale, University of Science and Technology of China, Hefei, Anhui 230026, China. [2] These authors contributed equally: Zhengxin Zhu, Zaichun Liu. ✉email: weichen1@ustc.edu.cn

To ameliorate the intermittent renewable energy resources, electrochemical energy storage devices have been constructed and deployed[1–3]. Lithium-ion battery (LIB) as a representative energy storage technology has achieved commercialization over 30 years[4,5]. Despite their high energy density, the power performances of LIBs are limited[6]. Electric double-layer capacitors (EDLCs) deliver fast charge/discharge capability, high specific power (up to $10 \, \mathrm{kW \, kg^{-1}}$), and long cycle life (e.g., millions of cycles), which allow them to complement batteries for high power applications[7–9]. However, EDLCs contain carbon-based electrodes that store electrical charges via ion adsorption at the electrode|electrolyte interface and no electrochemical reactions are involved. This charge storage mechanism is responsible for the EDLCs low energy content[10]. Driven by the need to increase the specific energy while maintaining the specific power, metal-ion capacitors with asymmetric configurations by hybridization of a battery-type electrode and an EDLC-type electrode have been considered a promising direction to explore[11,12].

In recent years, different metal-ion capacitors including lithium[13], sodium[14,15], potassium[16], and zinc[10,17,18] have been reported in the literature. For example, Tang and coworkers reported a zinc-ion capacitor (ZIC) through an integrated design of Zn metal negative electrode, activated carbon (AC) positive electrode, and non-aqueous electrolyte[17]. The authors reported the ZIC with specific energy and power of $53 \, \mathrm{Wh \, kg^{-1}}$ and $1725 \, \mathrm{W \, kg^{-1}}$ (based on the weight of active materials), respectively, and capacitance retention of 91% after 20,000 cycles at $2 \, \mathrm{A \, g^{-1}}$. Compared with non-aqueous organic-based electrolyte solutions, cost-effective aqueous electrolyte solutions with good ionic conductivity and high safety become important for practical applications like grid storage[19,20]. Accompanying the research on aqueous ZICs, it was found that the Zn metal negative electrode in aqueous electrolyte encountered numerous issues such as hydrogen evolution reaction, corrosion, and dendrite growth[21]. Even though these issues have been partially resolved by some efforts in designing new electrolytes and applying for surface protection, most of the reported research about aqueous ZICs disclose testing specific currents $\leq 20 \, \mathrm{A \, g^{-1}}$ and lifespans <100,000 cycles[22–32]. Thus, developing high-energy capacitors without sacrificing their lifespan and power performance is crucial for their deep penetration in energy storage applications.

Electrocatalytic hydrogen gas has been considered a promising electrode material for energy storage systems due to its abundant resources, the lightest molecular mass, fast kinetics, and low overpotential in terms of hydrogen evolution and oxidation reactions (HER/HOR)[33–35]. Moreover, the hydrogen gas electrode can operate very stably in electrolytes with a full range of pH (0–14), making it a pH-universal electrode[11]. Thus far, a series of hydrogen gas-based batteries have been successively reported, which exhibited good rate capability and cycle life[35–41]. For example, Chen and coworkers designed an iodine-hydrogen gas battery with fast charge/discharge rates and stable cycle life at different pH of the aqueous electrolyte[41]. However, most of the reported cathode materials displayed a limited rate and cycle life due to the occurrence of dissolution and accumulation reactions, which deteriorated the performance of hydrogen gas batteries. Therefore, it is highly desirable to the development of a capacitor by taking the distinct advantages of the electrocatalytic hydrogen gas as an electrode to improve the energy content without sacrificing the power and cycle life of the device.

Herein, we report a class of electrocatalytic hydrogen gas capacitors (EHGCs) using carbon-based EDLC positive electrodes and electrocatalytic hydrogen gas negative electrodes in pH-universal aqueous electrolytes. Pt/C catalyst was chosen as the $H_2$ negative electrode of EHGC, which is different from hydrogen

electrosorption-based cell systems that involve hydrogen storage in carbon materials[42]. Carbonaceous materials including AC and reduced graphene oxide (rGO) as EDLC electrode active materials were used as the positive electrodes. It is because both the negative electrode and positive electrode can work steadily in any aqueous solutions that the designed EHGCs are able to operate well in pH-universal electrolytes. Figure 1a schematically illustrates the construction and working mechanism of aqueous EHGC in $H_3PO_4$ acidic electrolytes. A transmission electron microscope (TEM) image of the Pt/C catalyst shows its nanometer nature and a scanning electron microscopy (SEM) image of the AC shows its micron meter particular characteristics (Fig. 1a). During the charging process, anions (including $H_2PO_4^-$ and $HPO_4^{2-}$) in the electrolyte move towards and adsorb onto the AC surface to form an electric double layer (EDL), while $H^+$ from the electrolyte moves towards the negative electrode and forms $H_2$ gas under the Pt/C catalyst. The discharge process is the reverse of the charging process, where the adsorbed anions desorb and migrate into the electrolyte on the positive electrode and $H_2$ gas is oxidized on the negative electrode. It is worth noting that during the initial state, EHGC with an open-circuit voltage of 0.45 V delivers a discharge capacity of $27 \, \mathrm{mAh \, g^{-1}}$ (Supplementary Fig. 1), suggesting that some anions might adsorb on the positive electrode and contribute to the initial discharge capacity[43]. The voltage gap of conventional EDLC is theoretically limited by HER and oxygen evolution reaction (OER). Although the voltage of EDLC can reach the thermodynamic water splitting potential of 1.23 V (at 25 °C and 1 atm), the working potential of the carbon positive electrode is only half of its maximum because of the symmetrical configuration. However, the voltage gap of the EHGC is only limited by OER. The working potential of the carbon positive electrode can exert the whole voltage of EHGC and reach 1.23 V ideally. Therefore, with an initial potential gap between the $H_2$ negative electrode and AC positive electrode, the EHGCs are able to exert the extra capacity of the AC positive electrode in comparison with the conventional EDLCs, as shown in Fig. 1b. The working principle of EHGC in neutral and alkaline electrolytes is in common with that of the acidic electrolyte except for the different adsorbed ions on the AC positive electrode. The constructed EHGC in an acidic electrolyte of 9 M $H_3PO_4$ at 25 °C exhibits a specific capacitance of $295 \, \mathrm{F \, g^{-1}}$ at $1 \, \mathrm{A \, g^{-1}}$, specific energy of $45 \, \mathrm{Wh \, kg^{-1}}$ (based on the mass of active materials of both electrodes), and fast charge/discharge rates up to $30 \, \mathrm{A \, g^{-1}}$ with a specific capacitance of $141 \, \mathrm{F \, g^{-1}}$ (specific capacity of $47 \, \mathrm{mAh \, g^{-1}}$). In addition, the EHGC tested in a pH-neutral electrolyte solution of 1 M phosphate buffer solution (PBS) enables capacitance retention of 85% (final specific capacitance of about $114 \, \mathrm{F \, g^{-1}}$) after 100,000 cycles at $10 \, \mathrm{A \, g^{-1}}$ and 25 °C. We also demonstrate the ability of EHGCs to operate in the full pH range (i.e., from 0 to 14) effectively and in a wide temperature range (i.e., from −70 to 60 °C).

## Results

**Electrochemical performance of the EHGCs in acidic electrolytes.** The electrochemical performances of aqueous EHGC were firstly investigated in 9 M $H_3PO_4$ acidic electrolytes as shown in Fig. 2. The cyclic voltammetry (CV) curves at $10 \, \mathrm{mV \, s^{-1}}$ show that both EHGC and EDLC present a quasi-rectangular shape (Fig. 2a), indicating the formation of an EDL during cycling[28]. The specific current of EHGC is larger than that of EDLC. This feature supports the assumption of an improved capacitive behavior of the EHGC. In the galvanostatic charge/discharge (GCD) measurement at a specific current of $1 \, \mathrm{A \, g^{-1}}$ in the voltage range from 0 to 1.2 V (Fig. 2b), both EHGC and EDLC exhibit typical EDL behavior with discharge-specific capacitances of $295 \, \mathrm{F \, g^{-1}}$ and $109 \, \mathrm{F \, g^{-1}}$,

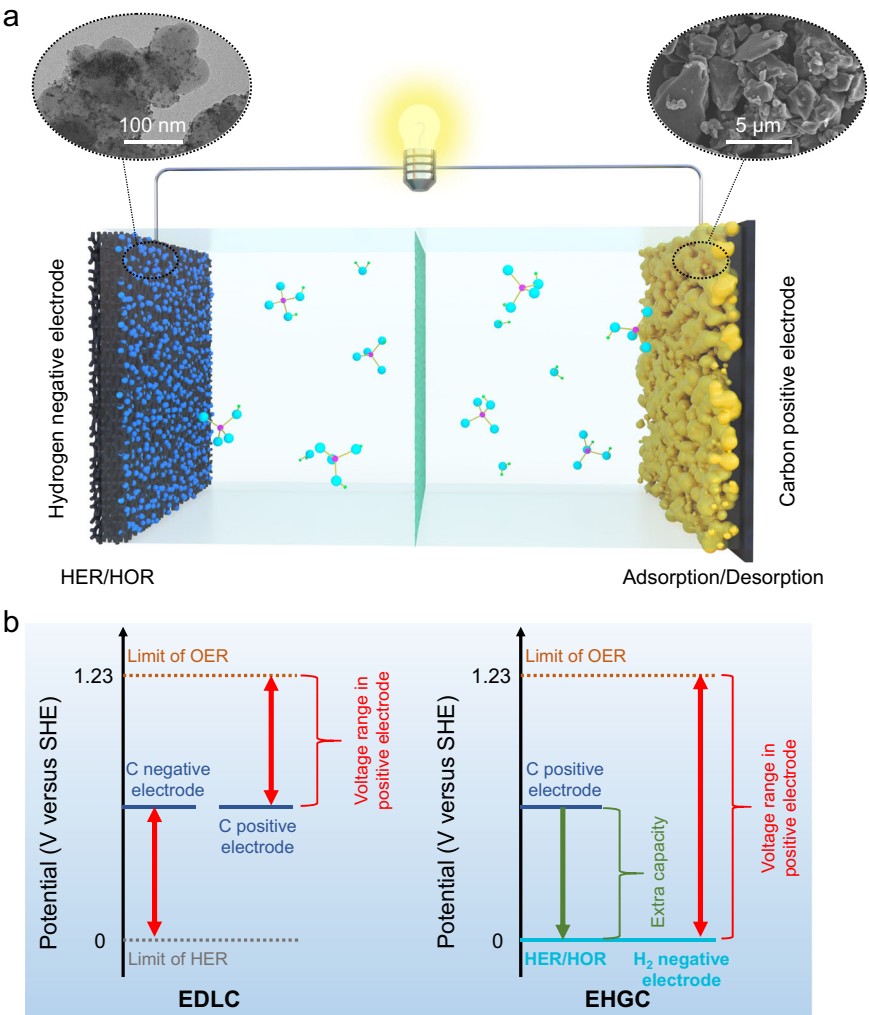

**Fig. 1 Schematic of the EHGC design. a** Schematic diagram showing the construction and working mechanism of the EHGC. In charge, the negative electrode occurs HER and the positive electrode occurs ions adsorption. In discharge, the negative electrode occurs HOR and the positive electrode occurs ions desorption. SEM image is AC (right) and TEM image is Pt/C (left), respectively. **b** Schematic diagram showing the advantage in capacitance of the EHGC in comparison with conventional EDLC. HER hydrogen evolution reaction. HOR hydrogen oxidation reaction, OER oxygen evolution reaction, EDLC electric double-layer capacitor, EHGC electrocatalytic hydrogen gas capacitor.

respectively, which is consistent with the CV results. When increasing the cell cut-off voltage, the slope of GCD curve tends to slacken with fast decreasing Coulombic efficiency (CE), which reveals that 0–1.2 V is the optimal electrochemical window in 9 M $H_3PO_4$ acidic electrolyte (Supplementary Fig. 2). In addition, EHGC delivers rate performance with the corresponding specific capacitances of 260, 190, 157, and 141 F g$^{-1}$ at 2, 10, 20, and 30 A g$^{-1}$, respectively (Fig. 2c and Supplementary Fig. 3). In contrast, the EDLC exhibits lower rate capability with the specific capacitances of 101, 86, 79, and 74 F g$^{-1}$ at 2, 10, 20, and 30 A g$^{-1}$, respectively (Supplementary Fig. 4). The comparison of EHGC and EDLC in specific capacitance and capacity is reported in Fig. 2d. At a specific current of 30 A g$^{-1}$, a specific capacitance of 141 F g$^{-1}$ (specific capacity of 47 mAh g$^{-1}$) was recorded in EHGC, which is about 1.9 times larger than EDLC with a specific capacitance of 74 F g$^{-1}$ (specific capacity of 25 mAh g$^{-1}$). Based on the mass of active materials of both electrodes, the EHGC shows the specific energy of 45 Wh kg$^{-1}$ at a specific power of 458 W kg$^{-1}$, which is about 4.5 times higher than EDLC with the specific energy of 10 Wh kg$^{-1}$ at a specific power of 267 W kg$^{-1}$. Moreover, the EHGC exhibits excellent cycling stability with stable capacitance after 25,000 cycles and slight loss (12.5%) after 50,000 cycles at a

specific current of 20 A g$^{-1}$ (Fig. 2e and Supplementary Fig. 5). In addition, the ex situ SEM, energy-dispersive X-ray (EDX), and X-ray photoelectron spectroscopy (XPS) measurements of the AC positive electrode (Supplementary Fig. 6) and Pt/C negative electrode (Supplementary Fig. 7) show no chemical or morphological changes after 10,000 cycles, which confirm the cycling stability of the EHGC. At a specific current of 4 A g$^{-1}$, the EHGC delivers an initial discharge capacitance of 220 F g$^{-1}$ and maintains capacitance retention of 86% after 5000 cycles (Supplementary Fig. 8). To verify the effect of the electrocatalytic $H_2$ gas electrode, we further tested the performance of the electrocatalytic capacitor in an Ar gas atmosphere under even higher AC loadings (8 mg cm$^{-2}$) on the positive electrode. It can be found that the electrocatalytic capacitor delivers a specific capacitance of 24 F g$^{-1}$ in Ar gas after 1000 cycle, where the EHGC in $H_2$ gas can reach a specific capacitance of 235 F g$^{-1}$ (Fig. 2f), indicating that $H_2$ gas is involved in the electrochemical reaction and significantly contribute capacitance to the EHGC. To highlight the electrochemical performance of EHGC, we have summarized and plotted the rate capacitance and cycling retention of the EHGC and various energy storage devices (Fig. 2g, h and Supplementary Table 1), such as ZICs[17,22,23,25,27,30], aqueous symmetric[44] and asymmetric[8,45,46] supercapacitors.

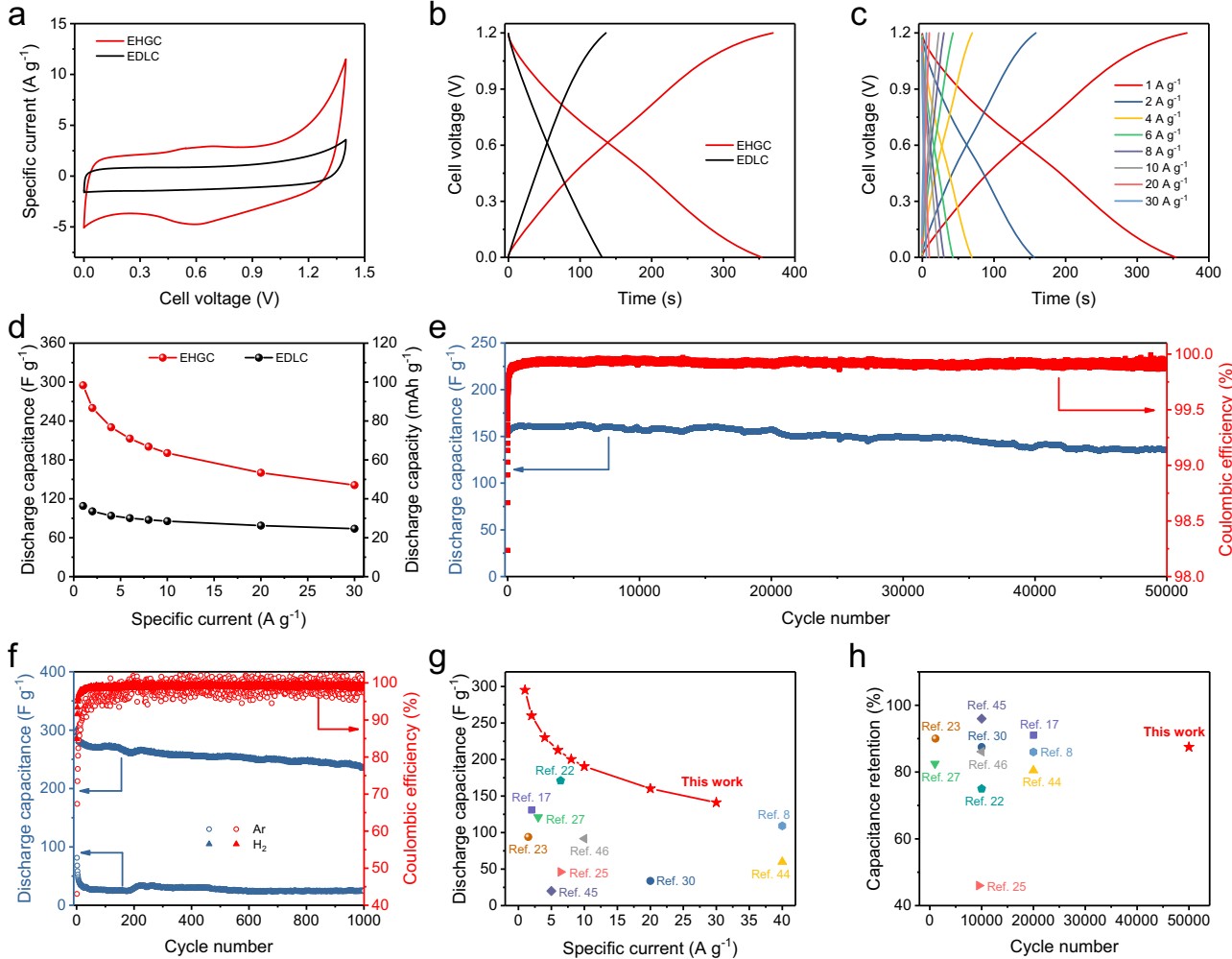

**Fig. 2 Electrochemical performance of the EHGCs in acidic electrolytes. a** CV curves of EHGC and EDLC at a scan rate of 10 mV s$^{-1}$. **b** Charge/discharge curves of EHGC and EDLC at a specific current of 1 A g$^{-1}$. **c** Charge/discharge curves at a voltage range of 0–1.2 V at different specific currents. **d** Specific capacitance (F g$^{-1}$) and specific capacity (mAh g$^{-1}$) as a function of the specific current of EHGC and EDLC. **e** Cycling performance at a specific current of 20 A g$^{-1}$. **f** Cycling performance under different gas environments (Ar and H$_2$) at a specific current of 1 A g$^{-1}$. The electrochemical measurements of the EDLCs and EHGCs were carried out at room temperature (25 °C) in an acidic electrolyte of 9 M H$_3$PO$_4$. The comparison of **g** rate capacitance and **h** cycling retention ratio between different supercapacitors and the EHGC. EDLC electric double-layer capacitor, EHGC electrocatalytic hydrogen gas capacitor.

**Electrochemical performance of the EHGCs in neutral electrolytes.** Figure 3a shows CV curves of the aqueous EHGC at various scan rates in a neutral electrolyte of PBS with a voltage range of 0–1.4 V. In addition, Fig. 3b and Supplementary Fig. 9 display that the neutral EHGC is able to work well in an optimal voltage range of 0–1.3 V with an optimal CE of 96.2%. A pair of redox peaks in the CV curve of EHGC are located at 0.55 and 0.41 V at a scan rate of 10 mV s$^{-1}$, which are originated from the oxygen functional groups of AC[24]. With increasing scan rates, the shapes of the CV curves and peak positions keep almost no change, which verifies the good rate capability of the electrocatalytic capacitor. As shown in Fig. 3c and Supplementary Fig. 10, the EHGC delivers discharge capacitances of 156, 134, and 116 F g$^{-1}$ at 4, 10, and 20 A g$^{-1}$, respectively. Figure 3d–f show the comparative performances between EHGC and EDLC. CV curves in Fig. 3d display that the specific current of the EHGC is larger than that of the EDLC at the same voltage (Fig. 3e), which is consistent with the result of GCD curves (190 F g$^{-1}$ for EHGC and 62 F g$^{-1}$ for EDLC at 1 A g$^{-1}$). In contrast, the EDLC exhibits similar charge/discharge rates but with lower capacitances (Supplementary Fig. 11). Figure 3f further highlights the advantage of the EHGC in capacitance and specific

capacity in comparison to the EDLCs. For example, the capacitance of the EHGC at a specific current of 1 A g$^{-1}$ (190 F g$^{-1}$) is about three times higher than that of the EDLC (62 F g$^{-1}$). Even at a specific current of 20 A g$^{-1}$, the capacitance of the EHGC (116 F g$^{-1}$) is about 2.6 times higher than that of the EDLC (45 F g$^{-1}$). The EHGC exhibits capacitance retention of 85% after 100,000 cycles at 10 A g$^{-1}$. It should be noticed that most of the capacitance is lost in the first 10,000 cycles (Fig. 3g). Moreover, this long-term cycling performance is well aligned with other power devices reported in the literature, such as metal-ion capacitors[17,25,27,28,30,31].

**Electrochemical performance of the EHGCs in alkaline electrolytes.** The electrochemical performance of the EHGC in 2 M KOH alkaline electrolyte was firstly investigated in different voltage windows. The GCD curves of the EHGC became asymmetric with CE of less than 96% when the cutoff voltage exceeded 1.1 V (Supplementary Fig. 12), in accordance with the result of CV (Fig. 4a). Accordingly, the electrochemical performance of the EHGC in the alkaline electrolyte was tested by GCD within the voltage range of 0–1.1 V. As shown in Fig. 4a,

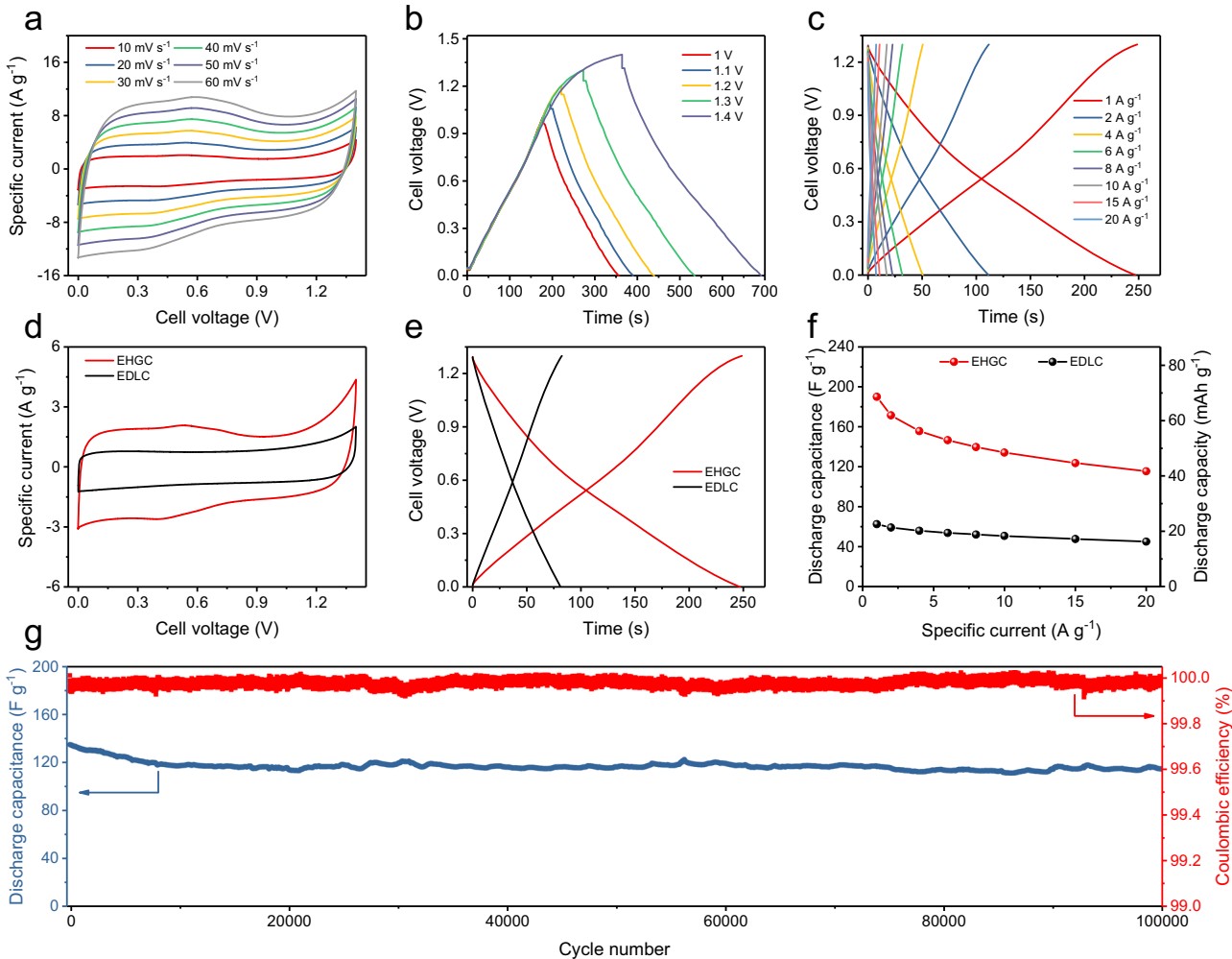

**Fig. 3 Electrochemical performance of the EHGCs in neutral electrolytes. a** CV curves at different scan rates. **b** Charge/discharge curves in different voltage ranges at a specific current of 1 A g$^{-1}$. **c** Charge/discharge curves of the EHGC in a voltage range of 0–1.3 V at different specific currents. **d** CV curves of EHGC and EDLC at a scan rate of 10 mV s$^{-1}$. **e** Charge/discharge curves of EHGC and EDLC at a specific current of 1 A g$^{-1}$. **f** Specific capacitance and specific capacity as a function of specific current for EHGC and EDLC. **g** Cycling performance of the EHGC at a specific current of 10 A g$^{-1}$. The electrochemical measurements of the EDLCs and EHGCs were carried out at room temperature (25 °C) in a neutral electrolyte of 1 M PBS. EDLC electric double-layer capacitor, EHGC electrocatalytic hydrogen gas capacitor.

the EHGC delivers a larger specific current and thus enclosed area than the EDLC, which reveals the higher specific capacitance. The GCD measurements were also performed at 1 A g$^{-1}$, showing that the EHGC can reach a better specific capacitance of 242 F g$^{-1}$ than that of the EDLC (86 F g$^{-1}$), which is an enhancement of as much as 280% (Fig. 4b). The GCD measurements of the EHGC at various specific currents ranging from 1 to 20 A g$^{-1}$ are shown in Fig. 4c and Supplementary Fig. 13. The specific capacitances calculated from the GCD curves at various specific currents show good rate capability, which delivers a capacitance of 224 F g$^{-1}$ at 2 A g$^{-1}$. When the specific current increases to 4, 6, 8, 10, 15, and 20 A g$^{-1}$, the capacitance can still keep as 209, 197, 189, 182, 164, and 146 F g$^{-1}$, respectively. As a comparison, the EDLC exhibits the lower specific capacitances of 84 F g$^{-1}$ at 2 A g$^{-1}$, and 72 F g$^{-1}$ at 20 A g$^{-1}$ (Supplementary Fig. 14). Comparisons in specific capacitance and specific capacity are also carried out at other specific currents, as summarized in Fig. 4d. The cycling stability of the EHGC was investigated by the GCD technique at 10 A g$^{-1}$, showing capacitance retention of 71% of the highest specific capacitance after 30,000 cycles (Fig. 4e).

**All-climate electrochemical performance of the EHGCs.** Considering the high thermal stability of electrode materials and the low freezing point (<−80 °C) of concentrated H$_3$PO$_4$ electrolyte (~9 M)[47], we tested the EHGC in a wide temperature range, i.e., from −70 to 60 °C. At a specific current of 40 A g$^{-1}$, the EHGC delivers a discharge-specific capacitance of 231 F g$^{-1}$ at a temperature of 60 °C (Fig. 5a). In addition, this EHGC performs a long lifespan with a capacity retention of 90% over 1000 cycles at 60 °C (Supplementary Fig. 15). The charge/discharge curves of the EHGC in the temperature of 25 and 40 °C show that the achieved capacity increases slightly with the operational temperature (Fig. 5b and Supplementary Fig. 16a). Afterward, we investigated the electrochemical performance of the EHGC at low temperatures from −20 to −50 °C (Fig. 5a and Supplementary Fig. 16b–d), displaying the discharge-specific capacitances of 139, 114, and 84 F g$^{-1}$ at −20, −40, and −50 °C at a specific current of 1 A g$^{-1}$, respectively, which is ~47, 39, and 28% of the capacitance at 25 °C, respectively. As a comparison, the capacitance of EHGC at 1 A g$^{-1}$ is 1.9 times higher than that of the EDLC with a capacitance of 48 F g$^{-1}$ at −20 °C. Meanwhile, the EHGC still keeps a better rate capability than the EDLC at −40 °C (Supplementary Fig. 17). When testing at

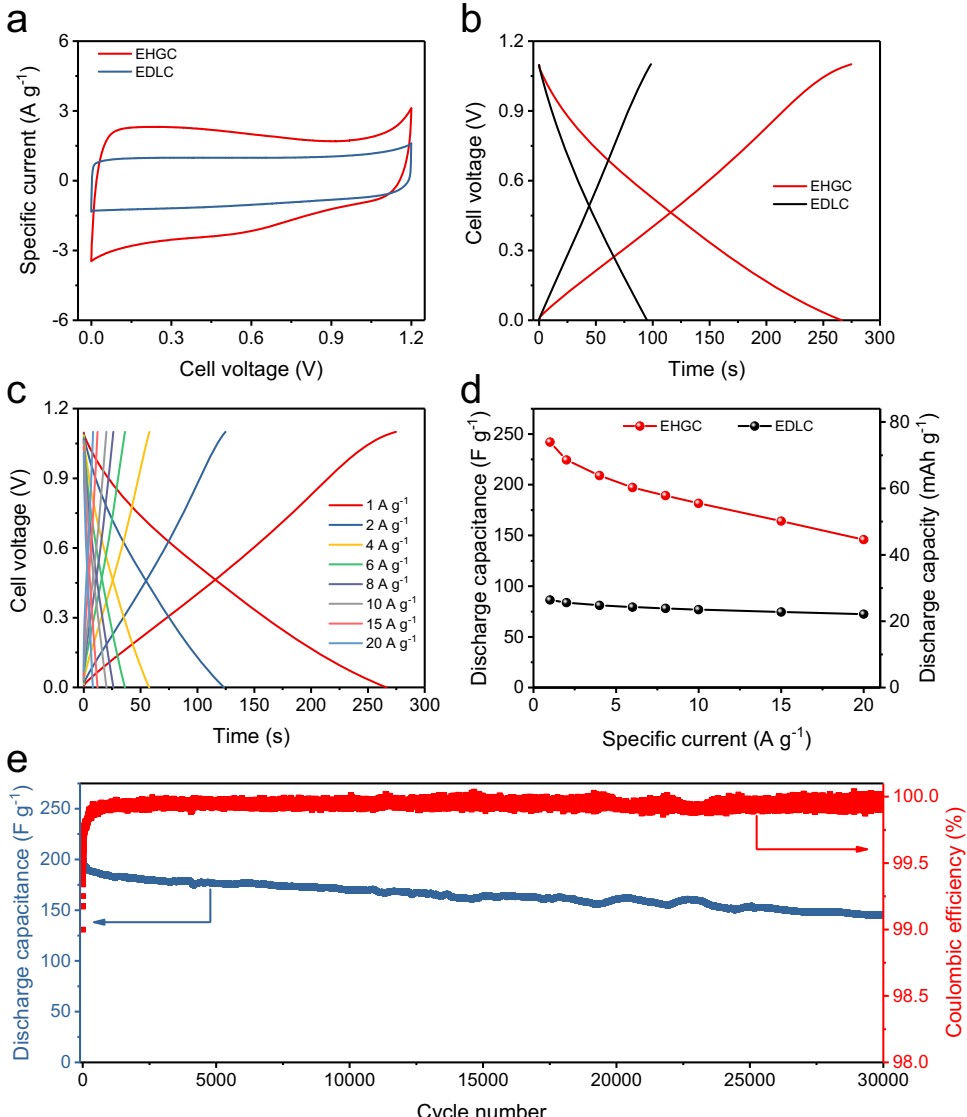

**Fig. 4 Electrochemical performance of the EHGCs in alkaline electrolytes. a** CV curves of EHGC and EDLC at a scan rate of 10 mV s⁻¹. **b** Charge/discharge curves of EHGC and EDLC at a specific current of 1 A g⁻¹. **c** Charge/discharge curves in a voltage range of 0–1.1 V at different specific currents. **d** Specific capacitance and specific capacity as a function of specific current for the EHGC and EDLC. **e** Cycling performance of the EHGC at a specific current of 10 A g⁻¹. The electrochemical measurements of the EDLCs and EHGCs were carried out at room temperature (25 °C) in an alkaline electrolyte of 2 M KOH. EDLC electric double-layer capacitor, EHGC electrocatalytic hydrogen gas capacitor.

−60 °C, the EHGC displays a specific capacitance of 108 F g⁻¹ at 0.05 A g⁻¹, and 51 F g⁻¹ at 0.5 A g⁻¹ (Fig. 5c). Even at a temperature of −70 °C, it also delivers a specific capacitance of 82 F g⁻¹ at 0.02 A g⁻¹ and 44 F g⁻¹ at 0.1 A g⁻¹ (Fig. 5d). The low-temperature rate capacitance performances shown by the EHGC are competitive compared with those of metal-ion capacitors reported in the literature and as summarized in Supplementary Table 2[48–50]. The comparisons in capacitance are also carried out with different specific currents at various temperatures, as summarized in Fig. 5e. Furthermore, long-term cycling stability at −20 °C is also demonstrated by the EHGC, showing stable capacitance for 10000 cycles at 4 A g⁻¹ with the CEs of ~100% (Fig. 5f).

**Suitability of the EHGC concept with other positive electrode active materials**. To demonstrate the applicability of the electrocatalytic capacitors, we further evaluate the EHGC by using reduced graphene oxide (rGO) as another positive electrode

material and coupling it with the H₂ gas negative electrode in all-pH electrolytes. The rGO was produced by a hydrothermal dehydration method to form dense agglomerates with layered structure as reported in the literature (Supplementary Fig. 18)[51]. As shown in Supplementary Fig. 19a, the CV of the EHGC clearly shows a pair of pronounced and broad redox peaks in the potential window of 0–1.2 V in an acidic electrolyte, which is indicative of pseudocapacitive behavior[51,52]. As a comparison, the EHGC exhibits a larger specific current and higher specific capacitance than the EDLC, as illustrated by CV and rate capacitances (Supplementary Fig. 19a–c). The GCD curves in Supplementary Fig. 19b show nearly symmetric triangular-shaped curves with distinct redox signatures around 0.6 V. In addition, the EHGC delivers a specific capacitance of 158 F g⁻¹ at 1 A g⁻¹ and 120 F g⁻¹ at 30 A g⁻¹, implying a good rate capability by using the rGO positive electrode. The long cycling test in Supplementary Fig. 19d shows good stability of the EHGC. Furthermore, the electrocatalytic capacitor was tested in neutral and

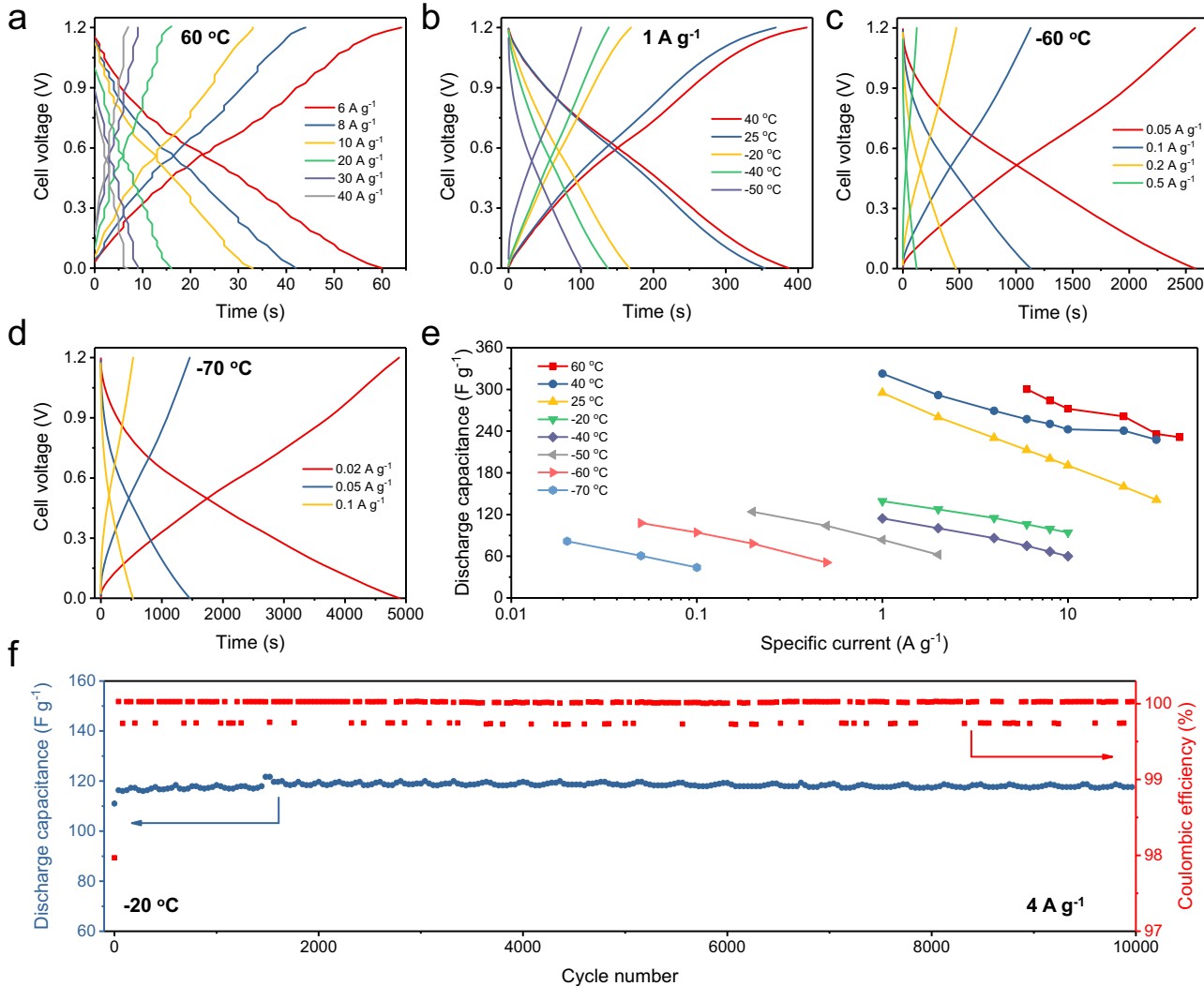

**Fig. 5 Electrochemical performance of the EHGCs in all climates. a** Charge/discharge curves with various specific currents at 60 °C. **b** Charge/discharge curves with a specific current of 1 A g$^{-1}$ at various operating temperatures from −50 to 40 °C. Charge/discharge curves with various specific currents at **c** −60 °C and **d** −70 °C. **e** Specific capacitance as a function of the temperature at various specific currents. **f** Cycling performance at 4 A g$^{-1}$ and −20 °C. The EHGCs were tested in the acidic electrolyte of 9 M H$_3$PO$_4$.

alkaline electrolytes, showing good rates, capacitances and cycle performances (Supplementary Figs. 20, 21), which confirms that it is feasible for the EHGC to operate with different EDLC electrode materials under all-pH conditions.

**Charge storage mechanisms and figures of merit of the EHGCs.**
To explore the different reaction mechanisms in EHGC and EDLC, density functional theory (DFT) calculations were performed to understand the HER/HOR activities in the negative electrodes of Pt and carbon. The calculation results indicate that the favorable H$^*$ adsorption kinetics of the Pt is better than the carbon components, as shown in Fig. 6a. Correspondingly, the HER/HOR occurs preferentially on the Pt rather than the carbon components, which is consistent with our experimentally obtained hydrogen gas electrode performance of Pt over the carbon electrode. In order to understand the reaction kinetics of the carbon negative electrode in different pH electrolytes, the adsorption energies of different anions in acidic (H$_2$PO$_4^-$) and alkaline (OH$^-$) electrolytes on the surface of carbon were also calculated. The relative adsorption energy (ΔE$_a$) values of H$_2$PO$_4^-$ and OH$^-$ on the carbon are −1.38 eV (Fig. 6b) and −1.06 eV

(Fig. 6c), respectively, which indicates that the ion adsorption on carbon is stronger in H$_2$PO$_4^-$ than OH$^-$. Bader charge analysis further demonstrated that the carbon positive electrode can accept more charge (0.50 |e|) from H$_2$PO$_4^-$ ions than OH$^-$ ions (0.41 |e|), indicating stronger interactions between the H$_2$PO$_4^-$ and carbon positive electrode. Combined with the adsorption energy results, it could be concluded that carbon as a positive electrode in the acidic electrolyte can deliver faster reaction kinetics than in an alkaline electrolyte.

Figure 6d summarizes the reduction potential versus pH of carbon positive electrode and H$_2$ gas negative electrode in a Pourbaix diagram. The media at which both the carbon positive electrode and H$_2$ gas negative electrode operate span a very wide pH range, from strongly acidic through neutral to strongly alkaline. The reduction potentials of the H$_2$ gas negative electrodes are pH-dependent and decrease by 59 mV per pH increment. The reduction potentials of carbon positive electrodes are mainly limited by the potentials of OER. Thus, due to the different potentials of HER and OER, the EHGC delivers different voltage windows in acidic, neutral, and alkaline electrolytes. The electrochemical performances of EHGCs in different pH electrolytes are summarized in Fig. 6e, showing that they deliver the

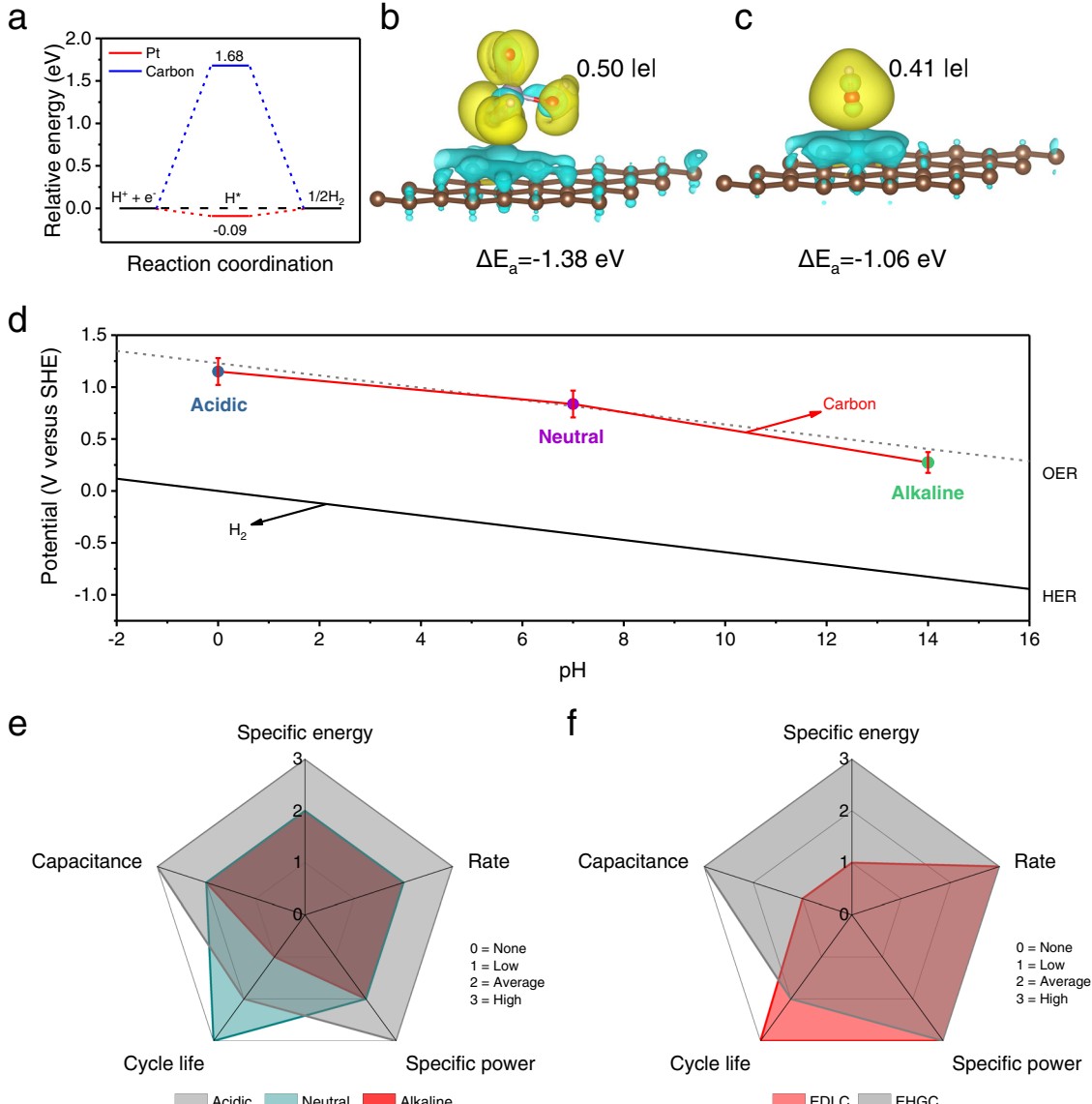

**Fig. 6 DFT calculations and comparisons of EHGCs with EDLCs in different pH electrolytes. a** Relative energy barriers for HER on the surfaces of Pt and carbon. The theoretical calculation models with the differential charge density of **b** $H_2PO_4^-$ and **c** $OH^-$ adsorption on carbon positive electrode.
**d** Correlation of potentials of the hydrogen gas negative electrode and carbon positive electrode with pH values of aqueous electrolytes. **e** Itemized comparison of three types of EHGCs in acidic, neutral, and alkaline electrolytes. **f** Itemized comparison between EHGC and EDLC in terms of capacitance, specific energy, specific power, rate capability, and cycle life. HER hydrogen evolution reaction, OER oxygen evolution reaction, EDLC electric double-layer capacitor, EHGC electrocatalytic hydrogen gas capacitor.

highest capacitance, specific energy, rate capability, and specific power in acidic electrolytes and the longest cycling capability in neutral electrolytes. These phenomena are ascribed to the fast reaction kinetics in acidic electrolytes and stable pH in neutral electrolytes. Furthermore, we compare some of the most important figures of merit for EHGC and EDLC, as displayed in Fig. 6f. The replacement of conventional carbon active materials with the $H_2$ gas at the negative electrode endows the EHGC with higher specific capacitance and specific energy while possessing high specific power and excellent rate capability over the EDLCs under the same conditions.

## Discussion

In this work, we have designed and tested a hybrid capacitive storage device named electrocatalytic hydrogen gas capacitor,

which was assembled by using electrocatalytic $H_2$ gas negative electrode and carbon-based positive electrodes. The EHGCs can operate in pH-universal electrolytes (i.e., from 0 to 14) and at a wide temperature range (i.e., from −70 to 60 °C). The EHGCs delivered a specific capacitance of 295 F g$^{-1}$ with a specific energy of 45 Wh kg$^{-1}$ at a specific power of 458 W kg$^{-1}$, which are about 4.5 times higher than that of the EDLCs. In addition, they exhibited good rate capability with a specific capacitance of 141 F g$^{-1}$ even at a specific current of 30 A g$^{-1}$ and a long cycle life of over 100,000 cycles, rendering them viable contenders toward practical energy storage applications. The electrochemical performances of the EHGCs are among the highest specific capacitance and longest cycle life ever reported for hybrid capacitors including ZICs and aqueous asymmetric supercapacitors. Furthermore, the theoretical calculations revealed the energy storage mechanism of the catalytic hydrogen gas negative

electrode over the carbon negative electrode. This work demonstrated a proof of concept for developing hybrid capacitive devices and showed the importance of the electrocatalytic $H_2$ negative electrode to the performance enhancement of the conventional EDLCs.

## Methods

**Materials**. The chemicals and materials in this work are commercially available and were used as received: Pt/C powder (20% Pt on Vulcan XC-72, Premetek), activated carbon (AC, YEC-8, Fuzhou Yihuan Carbon., Ltd), graphene oxide (GO) aqueous solution (5 mg mL$^{-1}$, Goographene), glass-fiber separator (GF-8, Whatman), polyvinylidene fluoride (PVDF, MTI), $N$-methyl-2-pyrrolidone (NMP, Aladdin), gas diffusion layer (GDL, Fuel Cell Store), $H_3PO_4$ (Sinopharm Chemical Reagent Co., Ltd., A.R.), 1 M potassium phosphate buffer solution (PBS, Sinopharm Chemical Reagent Co., Ltd., A.R., pH 7), KOH (Sinopharm Chemical Reagent Co., Ltd., A.R.), acetylene black (AB, MTI), titanium foil (thickness of 50 μm, 99.99%, MTI) and deionized water (resistance of 18.2 MΩ, Milli Q). In a typical synthesis of rGO, 75 mL of GO aqueous solution was sealed in a 100 mL Teflon-lined autoclave with air atmosphere and maintained at 180 °C for 6 h. The sample was cooled to ~25 °C, and then the resultant black product was filtered through centrifugation (TG16-WS, Cence Xiangyi) and washed using deionized water.

**Assembly of EHGCs**. Aqueous EHGCs were assembled by applying a plane-parallel electrode geometry in an $H_2$ gas-sealed Swagelok cell, with AC or rGO as positive electrode materials and Pt/C-catalyzed $H_2$ gas as negative electrode materials. The EHGC is designed in the pressurized Swagelok cell that is operated in the $H_2$-sealed condition, as shown in Supplementary Fig. 22[35,36]. Supplementary Fig. 22a shows each component of the device, where the stainless steel inlet and outlet valves are connected with Swagelok Klein Flange adapters (KF25). The positive and negative electrodes are sandwiched by a separator in a plane-parallel electrode geometry (Supplementary Fig. 22b), which is tightly placed on between two adapters where a polytetrafluoroethylene-centered O-ring is placed. Finally, the pressurized Swagelok cell is sealed with the assistance of a clamp and filled with high-purity hydrogen gas (Supplementary Fig. 22c). The glass-fiber separators located between negative and positive electrodes were wetted with the following electrolytes (40 μL): 9 M $H_3PO_4$ for acidic electrolyte; 1 M PBS (pH 7) for neutral electrolyte; 2 M KOH for alkaline electrolyte. For the selection of the acidic electrolyte, $H_3PO_4$ as a proton donor in the electrolyte is more moderate against the corrosion of stainless steel devices than the $H_2SO_4$ and HCl[47]. In addition, the high-concentration $H_3PO_4$ not only maintains the stabilization of pH but also keeps it in liquid states at low temperatures down to −80 °C[47]. Therefore, we chose the $H_3PO_4$ solution as the acidic electrolyte to test our EHGCs. For the selection of neutral electrolyte, the PBS is able to help stabilize the pH change, while other electrolyte such as $K_2SO_4$ is not[37,41]. Therefore, we chose PBS rather than $K_2SO_4$ solution as the electrolyte to stabilize the pH of the neutral system. For the selection of alkaline electrolytes, the KOH solution is the most commonly used electrolyte for supercapacitor in alkaline conditions. Therefore, we also chose the KOH solution as the alkaline electrolyte[39]. AC and Pt/C catalysts were commercially purchased. The carbon electrode slurry was made of AC or rGO, AB, and PVDF with a mass ratio of 8:1:1. Subsequently, the carbon electrodes were prepared by coating electrode slurry on titanium foil with a mass loading of 2 and 8 mg cm$^{-2}$. The negative electrode was mixed with Pt/C and PVDF with a mass ratio of 9:1. The mass loading of active materials on the negative electrode was ~0.2 mg cm$^{-2}$. The diameter of all electrodes is 12 mm.

**Electrochemical measurements**. CV tests were performed on a Bio-Logic VMP3 electrochemical station. GCD measurements were performed on a LANHE battery testing instrument or the Bio-Logic VMP3 electrochemical station. Unless otherwise specified, the testing temperature is ~25 °C. For low-temperature tests, the cells were placed in a Meiling ultralow temperature refrigerator from −20 to −70 °C and tested by a Neware battery testing instrument. For high-temperature tests, the cells were placed in an electric oven (Jing Hong) at 40 and 60 °C.

**Calculations of specific capacity, specific capacitance, specific energy, and specific power**. The specific capacity ($Q$, in the unit of mAh g$^{-1}$) and specific capacitance ($C$, in the unit of F g$^{-1}$) of the EHGCs and EDLCs were calculated from the discharge curves based on the following equations[24]:

$$Q = I * \Delta t/m \tag{1}$$

$$C = 3.6 * Q/U \tag{2}$$

where $I$ is the current, $\Delta t$ is the discharge time, $U$ is the testing voltage range, and $m$ is the mass of active material from the positive electrode. For example, when the specific current is 1 A g$^{-1}$ (mass from the positive electrode), the discharge time is 354 s, and the testing voltage range is 1.2 V (Fig. 2b), the specific capacity ($Q$) is calculated as 98.3 mAh g$^{-1}$ and the specific capacitance ($C$) is calculated as 295 F g$^{-1}$, according to the Eqs. 1 and 2.

The specific energy ($E$) and specific power ($P$) of the EHGCs and EDLCs were calculated based on the following equations[32]:

$$E = \int IV/M dt \tag{3}$$

$$P = E/\Delta t \tag{4}$$

where $\Delta t$ is the discharge time, $I$ is the current, $V$ is the voltage after Ohmic drop, and $M$ is the mass of active materials from the positive and negative electrodes. For example, when the specific current is 1 A g$^{-1}$ (mass from the positive electrodes), the mass of the positive electrode is 2 mg, the mass of negative electrode is 0.2 mg, and the discharge time is 354 s (Fig. 2b), the specific energy ($E$) is calculated as 45 Wh kg$^{-1}$, and specific power ($P$) is calculated as 458 W kg$^{-1}$, according to the Eq. 3 and 4.

**Physicochemical characterizations**. The morphologies and sizes of electrodes were characterized by the Hitachi8220 SEM instrument. EDX spectroscopy were also characterized by SEM analysis. The TEM images were acquired on a Hitachi HT-7700 transmission electron microscope. The Raman spectra were collected on a Raman spectrometer (LabRamHR Evolution), using laser excitation at 532 nm. The XPS measurements of the carbon positive electrode and Pt/C negative electrode were performed on ESCALAB 250Xi. The electrode harvesting, sampling, and transport were carried out in an air atmosphere without any additional treatment.

**Computational methods**. All calculations were carried out with the standard DFT using Vienna ab initio Simulation Package (5.4.4 VASP) within the generalized gradient approximation (GGA) as formulated by the Perdew–Burke–Ernzerhof (PBE) functional[53,54]. An energy cutoff of 500 eV was adopted, and the Brillouin zone was sampled with $3 \times 3 \times 1$ k-points using the Monkhorst-Pack scheme grid for geometry optimization and self-consistent calculations. A single layer of $4 \times 4 \times 1$ supercell in carbon supercells were selected to compare the adsorption of different anions. A single layer of $4 \times 4 \times 1$ supercell in graphite and two layers of Pt (111) loaded $4 \times 4 \times 1$ supercell carbon was selected to compare HER performance. A vacuum space exceeding 15 Å in the $z$-direction was employed to avoid the interaction from nearby layers. The atomic position was fully relaxed until the maximum force on each atom and total energy change was less than 0.02 eV/Å and $10^{-5}$ eV. The total energy was corrected with zero-point energy[55].

## Data availability

All data generated in this study are provided in the Source Data file and its Supplementary Information. Source data are provided with this paper.

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

## Acknowledgements

W.C. acknowledges the startup funds from USTC (Grant # KY2060000150) and the Fundamental Research Funds for the Central Universities (Grant # WK2060000040). We thank the support from USTC Center for Micro and Nanoscale Research and Fabrication. This research was also supported by the advanced computing resources provided by the Supercomputing Center of the USTC.

## Author contributions

Z.Z. and W.C. conceived the idea and directed the experiments. Z.L. provided the theoretical calculations. Z.Z. performed the experiments and characterizations. Y.Y. drew the schematic diagram. W.C. supervised the project. Z.Z., Z.L., Y.Y., Y.M., T.J., Q.P., W.W., and W.C. discussed the results. Z.Z. and W.C. drafted the manuscript.

## Competing interests

The authors declare no competing interests.
