## [Peer Review File · Nature Communications]

Reviewers' comments:

Reviewer #1 (Remarks to the Author):

I have read with quite attention the manuscript "pH-Universal Electrocatalytic Hydrogen Gas Capacitors" by Zhu et al. The authors have demonstrated a novel concept of electrocatalytic hydrogen gas capacitors (EHGCs) using hydrogen gas as anode and carbon as cathode capable of operating in acid, neutral and alkaline. The schematic of the device is proposed. The authors have tested the EHGCs in all three media through CVs, discharges/charges at different potential and stability tests. A deep characterization of the material is proposed supported by DFT analysis.

I find the results interesting and quite promising. However, I am expecting a deeper discussion on why the authors have selected those electrolytes (H₃PO₄, PBS and KOH) and no other electrolytes, what are the phenomena occurring inside. Deeper scientific discussion on explaining the results is absolutely needed. It looks to me that only results (quite good) are reported without explaining them properly.

The authors are missing an important part related to the separator.

The discussion on the separator is missing. The separator is crucial because it needs to physically separate physically hydrogen from the two electrodes. In a fuel cell (H₂/air) in fact the OCV is 1 V using Pt, I guess in this case using AC and Pt should be 0.7. The authors will have a lower potential due to the mixed potentials created with the hydrogen moving from one side to the other. Separator discussion has to be improved.

I am missing details related to the device fabrication which is crucial for other scientists around the world to replicate the device and the results.

The authors are showing ideal situation in Figure 1.B. However, knowing that HER and OER are problematic and carbon are not very efficient electrocatalysts for HER and OER, the window potential can be increased up to 1.6-1.8 V without problems. AC are very bad catalysts for HER and OER in acid media. In alkaline media instead, due to their high surface area, AC has lower overpotentials compared to acid media.

The CVs done in acid, neutral and alkaline media show that it is not a rectangular shape, but some reactions are occurring in the EHGC. The authors should explain why there are reactions occurring and which reactions are happening.

It is not that clear when it was used AC or rGO. It should be clearly marked within the text and the figures.

It is very interesting the part related to low temperature analysis.

Reviewer #2 (Remarks to the Author):

Authors have reported on the pH-universal electrocatalytic hydrogen gas capacitors. The work itself is interesting and brings some new insights into electrochemical capacitors development, however, the recent version requires serious corrections before considering for further steps of publication.

1. Authors should refrain from using 'electrochemical double layer capacitors term. The correct terms

are 'electrochemical capacitors' or 'electric double-layer capacitors'.

2. All the numbers that concerns capacitance values given in the manuscript should be expressed as integers. Having so small mass loadings in the cells, expression with so high precision has no meaning.

3. Authors should clearly point out the difference between the system developed by them and the systems already developed in the past by other teams and using hydrogen electrosorption (see the works of Deyang Qu, Francois Beguin, Elzbieta Frackowiak, Krzysztof Jurewicz, Encarnacion Raymundo-Pinero from early 2000'). It seems that the concept is the same (storage of carbon near HEP).

4. Authors should provide typical three-electrode measurements to prove the operation principles and potential plateau on the negative electrode. Here, please be aware that the terms cathode and anode are incorrect in the electrochemical capacitors, as there are no redox processes in the system (at least by definition).

5. Authors should explain why they express the values per mass of cathode (L343) only? In the asymmetric systems, total mass should be considered (preferably with electrolyte).

6. Regarding the acidic electrolyte - how did the Authors avoid the corrosion?

7. Coulombic efficiency in such systems is misleading. Please provide energetic efficiency and calculate the capacitance from energy of the cell or using constant power discharge.

8. In order to show that the concept works as energy storage system, self-discharge and leakage currents should be presented.

9. THE CAPACITANCE OF CAPACITORS IS CONSTANT REGARDLESS VOLTAGE. How to understand the profiles of voltage vs. capacitance???? Authors should look at CV profiles and then re-consider their graphs.

10. What is the benefit of the system if the voltage is limited to 1.2 V?

11. What are the capacitance values of single electrodes? And how these electrodes were balanced? Having 50 Wh/kg at 1.2 V it seems that the capacitance reaches 1000 F/g (~330 mAh/g).

For all these reasons, I cannot recommend the paper for publishing in the present form, I recommend rejection and re-submission once seriously and correctly revised.

Response to Reviewers' Comments

Reviewer #1 (Remarks to the Author):

General Comment: I have read with quite attention the manuscript “pH-Universal Electrocatalytic Hydrogen Gas Capacitors” by Zhu et al. The authors have demonstrated a novel concept of electrocatalytic hydrogen gas capacitors (EHGCs) using hydrogen gas as anode and carbon as cathode capable of operating in acid, neutral and alkaline. The schematic of the device is proposed. The authors have tested the EHGCs in all three media through CVs, discharges/charges at different potential and stability tests. A deep characterization of the material is proposed supported by DFT analysis.

Our response: We gratefully appreciate the reviewer’s careful reading and positive comments to our manuscript. The manuscript has been thoroughly revised according to the reviewer’s insightful suggestions as detailed below. All changes have been highlighted in red in the revised manuscript.

Comment 1: I find the results interesting and quite promising. However, I am expecting a deeper discussion on why the authors have selected those electrolytes (H_3PO_4 , PBS and KOH) and no other electrolytes, what are the phenomena occurring inside. Deeper scientific discussion on explaining the results is absolutely needed. It looks to me that only results (quite good) are reported without explaining them properly.

Our response: Thanks for the reviewer’s valuable comment. We would like to point out that the pH of our EHGCs changes with the cell charge and discharge. The pH increases in the charge process and decreases in the discharge process. However, it returns back to the original pH value after the fully charge or discharge. According to the Nernst equation ($E = -0.0592 \cdot \text{pH}$), the pH change will impact cell voltage by shifting the potential of the H_2 gas negative electrode. For the selection of acidic electrolyte, H_3PO_4 as a proton donor in the electrolyte is more moderate against the corrosion of stainless-steel device than the typical H_2SO_4 and HCl in our experiments. In addition, through our previous work (J. Am. Chem. Soc. 2021, 143, 20302-20308), the high-concentration H_3PO_4 not only maintains the stabilization of pH, but also keeps in liquid state at low temperatures down to $-80\text{ }^\circ\text{C}$. Therefore, we chose the H_3PO_4 solution as the acidic electrolyte to test our EHGCs. For the selection of neutral electrolyte, the PBS is able to help stabilize the pH change, while other

electrolyte such as K_2SO_4 is not. Therefore, we chose the PBS rather than conventional K_2SO_4 solution as the electrolyte to stabilize the pH of the neutral system. For the selection of alkaline electrolyte, the KOH solution is the most commonly used electrolyte for supercapacitor in the alkaline conditions. Therefore, we also chose the KOH solution as the alkaline electrolyte.

Our revision: According to the reviewer's comment, we made changes to the revised manuscript as shown in the following. In the revised manuscript, we added the discussion as follows: *“For the selection of acidic electrolyte, H_3PO_4 as a proton donor in the electrolyte is more moderate against the corrosion of stainless-steel device than the typical H_2SO_4 and HCl in our experiments. In addition, the high-concentration H_3PO_4 not only maintains the stabilization of pH, but also keeps in liquid state at low temperatures down to $-80\text{ }^\circ\text{C}$. Therefore, we chose the H_3PO_4 solution as the acidic electrolyte to test our EHGCs. For the selection of neutral electrolyte, the PBS is able to help stabilize the pH change, while other electrolyte such as K_2SO_4 is not. Therefore, we chose the PBS rather than conventional K_2SO_4 solution as the electrolyte to stabilize the pH of the neutral system. For the selection of alkaline electrolyte, the KOH solution is the most commonly used electrolyte for supercapacitor in the alkaline conditions. Therefore, we also chose the KOH solution as the alkaline electrolyte.”*

Please see the highlighted part in red color on page 20 in the revised manuscript.

Comment 2: The authors are missing an important part related to the separator. The discussion on the separator is missing. The separator is crucial because it needs to physically separate physically hydrogen from the two electrodes. In a fuel cell (H_2 /air) in fact the OCV is 1 V using Pt, I guess in this case using AC and Pt should be 0.7. The authors will have a lower potential due to the mixed potentials created with the hydrogen moving from one side to the other. Separator discussion has to be improved.

Our response: Thanks for the reviewer's good comment and kind suggestion. We agree with the reviewer's comment that the separator plays an important role in the battery cell. However, we would like to point out that the proposed EHGC uses ordinary glass fiber separator, which plays a role in conducting ions and separating the cathode and anode. Different from a fuel cell (H_2 /air), where H_2 and O_2 need to be physically separated, only H_2 is presented in our EHGC. Therefore, there is no need to physically separate the H_2 from the two electrodes. In fact, H_2 that is sealed in the Swagelok cell only reacts on the negative electrode, which is demonstrated in many previous rechargeable H_2 batteries and showed good electrochemical

performance (Nat. Energy 2018, 3, 428-435; Proc. Natl. Acad. Sci. USA 2018, 115, 11694-11699; Nano Lett. 2020, 20, 3278-3283; Adv. Funct. Mater. 2021, 31, 2101024; J. Am. Chem. Soc. 2021, 143, 20302-20308). In addition, the H₂ in the cell does not affect ions adsorption/desorption of the AC positive electrode during cell charge/discharge. Therefore, cost-effective glass fiber separator was used in our EHGCs.

Our revision: According to the reviewer's comment, we made changes to the revised manuscript as shown in the following. In the revised manuscript, we added the content as follows: "*The regular glass-fiber separators located between negative electrode and positive electrode were wetted with the following electrolytes (40 μL)*" Please see the highlighted part in red color on page 20 in the revised manuscript.

Comment 3: I am missing details related to the device fabrication which is crucial for other scientists around the world to replicate the device and the results.

Our response: Thanks for the reviewer's good comment. The EHGC and EDLC are all designed in a pressurized Swagelok cell that is operated in the sealed condition as shown in Fig. R1, which is consistent with our previous work (Nat. Energy 2018, 3, 428-435; Proc. Natl. Acad. Sci. USA 2018, 115, 11694-11699; Nano Lett. 2020, 20, 3278-3283; Adv. Funct. Mater. 2021, 31, 2101024; J. Am. Chem. Soc. 2021, 143, 20302-20308). Fig. R1a shows each component of the device, where the stainless-steel inlet and outlet valves are connected with Swagelok Klein Flange adapters. The positive and negative electrodes are sandwiched by a separator with a typical coin-cell stack (Fig. R1c), which is tightly placed on between two adapters where a polytetrafluoroethylene-centered O-ring is placed. Finally, the pressurized Swagelok cell is sealed with the assistance of a clamp and filled with high-purity hydrogen gas (Fig. R1b).

Fig. R1. The setup of the Swagelok-type EHGC cell.

Our revision: According to the reviewer's comment, we made changes to the revised manuscript and supporting information as shown in the following. In the revised manuscript, we added the discussion as follows: *“The EHGC is designed in the pressurized Swagelok cell that is operated in the H₂-sealed condition, as shown in Supplementary Fig. 19, which is consistent with our previous work^{33,34}. Supplementary Fig. 19a shows each component of the device, where the stainless-steel inlet and outlet valves are connected with Swagelok Klein Flange adapters. The positive and negative electrodes are sandwiched by a separator with a typical coin-cell stack (Supplementary Fig. 19c), which is tightly placed on between two adapters where a polytetrafluoroethylene-centered O-ring is placed. Finally, the pressurized Swagelok cell is sealed with the assistance of a clamp and filled with high-purity hydrogen gas (Supplementary Fig. 19b).”* Please see the highlighted part in red color on page 20 in the revised manuscript and Supplementary Fig. 19 in the revised supporting information.

Comment 4: The authors are showing ideal situation in Figure 1.B. However, knowing that HER and OER are problematic and carbon are not very efficient electrocatalysts for HER and OER, the window potential can be increased up to 1.6-1.8 V without problems. AC are very bad catalysts for HER and OER in acid media. In alkaline media instead, due to their high surface area, AC has lower overpotentials compared to acid media.

Our response: Thanks a lot for the reviewer's insightful comment. We agree with the reviewer's comment that carbon are not very efficient electrocatalysts for HER and OER with the possible window potential up to 1.6-1.8 V. In this regard, we firstly want to point out that Fig. 1b is only the schematic diagram showing the ideal limited potential between HER and OER in the EDLC and EHGC without considering of their overpotentials. The aim of the Fig. 1b is to illustrate the much larger voltage range of the positive electrode in the EHGC than the EDLC, thus higher capacity can be obtained in the EHGC. Secondly, our practical AC materials with many functional groups on the surface can be oxidized before the potential of OER. Thus, AC electrode usually delivers a narrower voltage window in practical situation, which is consistent with previous literature (Adv. Energy Mater. 2020, 10, 2001705; Chem. Eur. J. 2018, 24, 3639-3649). Therefore, the proposed EHGC delivers a voltage range of 1.1-1.3 V in practice.

Comment 5: The CVs done in acid, neutral and alkaline media show that it is not a rectangular shape, but some reactions are occurring in the EHGC. The authors should explain why there are reactions occurring and which reactions are happening.

Our response: Thanks for the reviewer's valuable comment. The cathodic and anodic peaks in CVs can be ascribed to the redox reactions of EHGC, which are originated from the oxygen functional groups of AC, which is consistent with previous literature (Adv. Energy Mater. 2020, 10, 2001705; Energy Storage Mater. 2021, 42, 705-714). In order to confirm this behavior, we have conducted three-electrode tests to the AC positive electrode in 9 M H₃PO₄ electrolyte. As shown in Fig. R2, we can also see that the CV curve of the AC electrode is not in a rectangular shape due to the surface oxygen functional groups induced redox peaks. Therefore, the CVs done in the acidic, neutral and alkaline media show that it is not a rectangular shape.

Fig. R2. CV curve of AC positive electrode at a current density of 1 A g⁻¹ in the 9 M H₃PO₄ electrolyte.

Our revision: According to the reviewer's comment, we made changes to the revised manuscript as shown in the following. In the revised manuscript, we added the discussion as follows: *“A pair of redox peaks in CV curve of EHGC are located at 0.55 V and 0.41 V at a scan rate of 10 mV s⁻¹, which are originated from the oxygen functional groups of AC²².”* Please see the highlighted part in red color on page 9 and page 10 in the revised manuscript.

Comment 6: It is not that clear when it was used AC or rGO. It should be clearly marked within the text and the figures.

Our response: Thanks a lot for the reviewer's careful reading and insightful suggestion. We have clearly marked the words of AC or rGO in the figures when the AC or rGO was used as the electrode of EHGC.

Comment 7: It is very interesting the part related to low temperature analysis.

Our response: We gratefully appreciate the reviewer's positive comment. In order to highlight the unique advantages of our designed EHGC in different temperatures, we have tested it at higher temperatures (40 °C and 60 °C) and lower temperatures (−50 °C, −60 °C, and −70 °C), which are critical to indicate the all-climate accommodative performance. Meanwhile, we have added a section of “All-climate electrochemical performance of the EHGCs” to discuss the EHGC under different temperatures. New figures of Fig. 5 and Supplementary Fig. 12-14 have been added to the revised manuscript and supplementary information. The new contents of Fig.5 are shown in the following:

Considering the high thermal stability of electrode materials and the low freezing point of concentrated H₃PO₄ electrolyte, we speculated that the EHGC can be operated in a wide temperature range. Our previous research has demonstrated the low-temperature feasibility of concentrated H₃PO₄ electrolytes in rechargeable hydrogen gas-proton batteries, which can be applied to our EHGCs for low-temperature applications⁴⁵. The charge/discharge curves of the EHGC in the temperature range from 25 to 40 °C show that the achieved capacity increases slightly with the operational temperature (Fig. 5a and Supplementary Fig. 12a). Surprisingly, the EHGC displays an excellent rate of 40 A g⁻¹ with a high capacitance of 231 F g⁻¹, at the higher temperature of 60 °C (Fig. 5b). In addition, this EHGC performs a long lifespan with a high capacity retention of 90% over 1000 cycles at 60 °C (Supplementary Fig. 13). Afterwards, we investigated the electrochemical performance of the EHGC at low temperatures from −20 to −50 °C (Fig. 5a and Supplementary Fig. 12b-d), displaying the discharge capacities of 139, 114, and 84 F g⁻¹ at −20, −40, and −50 °C at a current density of 1 A g⁻¹, respectively, which is approximately 47%, 39%, 28% of the capacitance at 25 °C, respectively. As a comparison, the capacitance of EHGC at 1 A g⁻¹ is 1.9 times higher than that of the EDLC with a low capacitance of 48 F g⁻¹ at −20 °C. Meanwhile, the EHGC still keeps the higher rate capacitances than the EDLC at −40 °C (Supplementary Fig. 14). When testing at −60 °C, the EHGC displays a high capacitance of 108 F g⁻¹ at 0.05 A g⁻¹, and 51 F g⁻¹ at 0.5 A g⁻¹ (Fig. 5c). Even at an ultralow temperature of −70 °C, it also delivers a high capacitance of 82 F g⁻¹ at 0.02 A g⁻¹ and 44 F g⁻¹ at 0.1 A g⁻¹ (Fig. 5d), demonstrating the

superior rate capability of the EHGC under ultralow temperature conditions. To the best of our knowledge, such excellent rate capacitances of our EHGC at very low temperatures are among the best values reported so far in metal ion capacitors, as summarized in Supplementary Table 2⁴⁶⁻⁴⁸. Moreover, the comparisons in capacitance and specific capacity are also conducted with different current densities at various temperatures, as summarized in Fig. 5e, which delivers excellent temperature-tolerant properties of our EHGC. Furthermore, a long-term cycling stability at $-20\text{ }^{\circ}\text{C}$ is also manifested for the EHGC, showing no capacitance decay after 10000 cycles at 4 A g^{-1} with the CEs of approximately 100% (Fig. 5f).

Fig. 5. Electrochemical performance of the EHGCs in all climates. a Charge/discharge curves with a current density of 1 A g^{-1} at various operating temperatures from $-50\text{ }^{\circ}\text{C}$ to $40\text{ }^{\circ}\text{C}$. Charge/discharge curves with various current densities at b $60\text{ }^{\circ}\text{C}$, c $-60\text{ }^{\circ}\text{C}$, and d $-70\text{ }^{\circ}\text{C}$. e Capacitance and specific capacity as a function of the temperature at various current densities. f Cycling performance at 4 A g^{-1} and $-20\text{ }^{\circ}\text{C}$. The EHGCs were tested in the acidic electrolyte of $9\text{ M H}_3\text{PO}_4$.

Reviewer #2 (Remarks to the Author):

General Comment: Authors have reported on the pH-universal electrocatalytic hydrogen gas capacitors. The work itself is interesting and brings some new insights into electrochemical capacitors development, however, the recent version requires serious corrections before considering for further steps of publication.

Our response: We are very grateful to the reviewer's comments and suggestions, which are beneficial to improving the quality of our manuscript. We also would like to thank the reviewer for his/her acknowledgment of the new development of our EHGC. Meanwhile, we appreciate the reviewer for pointing out some problems, which are all very important for revising our manuscript. We have tried our best to conduct additional experiments and provide further discussions to address the comments of the reviewer. All changes have been highlighted in red color in the revised manuscript. We hope that the revised manuscript is suitable for publication in *Nature Communications*.

Comment 1: Authors should refrain from using 'electrochemical double layer capacitors term. The correct terms are 'electrochemical capacitors' or 'electric double-layer capacitors'.

Our response: Thanks a lot for the reviewer's comment. We have corrected all the expression term from "electrochemical double layer capacitors" to "electric double-layer capacitors".

Comment 2: All the numbers that concerns capacitance values given in the manuscript should be expressed as integers. Having so small mass loadings in the cells, expression with so high precision has no meaning.

Our response: Thanks a lot for the reviewer's insightful suggestion. We agree with the reviewer's comment that all the numbers that concerns capacitance values given in the manuscript should be expressed as integers. In this regard, we have changed the capacitance values from fractions to integers.

Comment 3: Authors should clearly point out the difference between the system developed by them and the systems already developed in the past by other teams and using hydrogen electrosorption (see the works of Deyang Qu, Francois Beguin, Elzbieta Frackowiak, Krzysztof Jurewicz, Encarnacion Raymundo-Pinero from early 2000'). It seems that the concept is the same (storage of carbon near HEP).

Our response: Thanks a lot for the reviewer's careful reading and constructive comment. Our newly developed EHGC is fundamentally different from all the previously reported capacitor systems including the mentioned hydrogen electrosorption systems. The difference and advancements of our work over the previous work of hydrogen electrosorption are shown in the following. First, the materials used in our EHGC and the working mechanism are different from the hydrogen electrosorption systems. Our designed EHGCs use hydrogen gas as the negative electrode, which is driven by the electrocatalytic hydrogen evolution and oxidation reactions, by involving the gas-liquid-solid three phase interfacial reactions of hydrogen gas under the Pt/C catalyst. However, the reported hydrogen electrosorption-based cell is based on hydrogen storage in carbon materials with high surface area as the negative electrode, where the capacity is limited by the capacity of the hydrogen gas that is physically stored in the carbon materials. While in our EHGC, the capacity of the negative electrode based on HER/HOR is much higher (theoretical capacity of 2976 mAh g^{-1} based on the mass of H_2O and the reaction of $2\text{H}_2\text{O} + 2\text{e}^- \leftrightarrow \text{H}_2 + 2\text{OH}^-$). Overall, our work provides an innovative approach for the design of EHGC for future development of advanced hydrogen gas-based cell systems.

Our revision: According to the reviewer's comment, we have already revised the manuscript. The detailed revision is shown as follows: "*Pt/C as a typical catalyst was chosen as the H_2 negative electrode of EHGC, which is different from hydrogen electrosorption-based cell systems that involve in the hydrogen storage in carbon materials⁴⁰.*" Please see the highlighted part with red background on page 4 in the revised manuscript.

Comment 4: Authors should provide typical three-electrode measurements to prove the operation principles and potential plateau on the negative electrode. Here, please be aware that the terms cathode and anode are incorrect in the electrochemical capacitors, as there are no redox processes in the system (at least by definition).

Our response: We appreciate the reviewer's insightful comment. To prove the operation principles and potential plateau on the negative electrode, we have conducted three-electrode measurements by using Pt/C-GDL as the working electrode, graphite rod as the counter electrode and Ag/AgCl as the reference electrode in different electrolytes (9 M H_3PO_4 , 1 M PBS, and 2 M KOH). High-purity hydrogen gas has been purged into the electrolytes on the surface of GDL. As shown in Fig. R1, all polarization curves display negligible overpotentials

for hydrogen evolution and oxidation reactions (HER/HOR), in which the potential linearly changes with current, suggesting that the polarization of the HER/HOR is mainly contributed by its electrical resistance. As an example of the cell in the electrolyte of 9 M H₃PO₄ (Fig. R1a), the overpotential at 5 mA cm⁻² is as low as 19 mV vs RHE in the HER region. In the HOR region, the overpotential at 5 mA cm⁻² is as low as 20 mV vs RHE. Therefore, it is well confirmed that the negative electrode delivers very low polarization when using as the H₂ electrode with HER/HOR by the excellent Pt/C bifunctional catalyst. Finally, we appreciate the reviewer's insightful suggestion on the terms of "anode and cathode". We have corrected the terms in our revised manuscript from "anode and cathode" to "negative electrode and positive electrode", respectively.

Fig. R1. HER/HOR polarization curves of the Pt/C electrode at a scan rate of 10 mV s⁻¹ in H₂-saturated (a) 9 M H₃PO₄ electrolyte, (b) 1 M PBS electrolyte, and (c) 2 M KOH electrolyte.

Comment 5: Authors should explain why they express the values per mass of cathode (L343) only? In the asymmetric systems, total mass should be considered (preferably with electrolyte).

Our response: Thanks a lot for the reviewer's comment and suggestion. The capacitance was initially calculated by the mass loading of positive electrode, which is because the hydrogen gas negative electrode with a very low mass loading (0.2 mg cm⁻²) of catalyst can provide much larger capacity (theoretical capacity of 2976 mAh g⁻¹ based on the mass of H₂O and the reaction of 2H₂O + 2e⁻ ↔ H₂ + 2OH⁻) than the positive AC electrode. However, we agree with the reviewer's comment that total mass of electrodes should be considered in the asymmetric systems when calculating the energy density and power density of EHGCs. Accordingly, we recalculate the values of energy density and power density based on the mass loading of the positive and negative electrodes. The EHGCs still deliver a high energy density of 45 Wh kg⁻¹ and a power density of 11 kW kg⁻¹, when considering the mass loading of 2

mg cm⁻² in the positive electrode and 0.2 mg cm⁻² in the negative electrode. We have changed the values of energy density and power density in the revised manuscript.

Our revision: According to the reviewer's comment, we made changes to the revised manuscript as shown in the following. The detailed revision is shown as follows: "*m is the mass of active materials from the positive and negative electrodes*" Please see the highlighted part in red color on page 21 in the revised manuscript.

Comment 6: Regarding the acidic electrolyte - how did the Authors avoid the corrosion?

Our response: Thanks for the reviewer's valuable comment. The corrosion is a common issue in the acidic electrolyte. However, we chose the H₃PO₄ as the acidic electrolyte, which is more moderate against the corrosion of stainless-steel device than the general H₂SO₄ and HCl throughout our experiments. The H₃PO₄ electrolyte without corrosion was also demonstrated by our previous work (J. Am. Chem. Soc. 2021, 143, 20302-20308). In addition, to avoid the possible corrosion of stainless-steel, a titanium foil was used as the collector current when we assembled the EHGCs in the acidic electrolyte. This information is also included in our revised manuscript.

Comment 7: Coulombic efficiency in such systems is misleading. Please provide energetic efficiency and calculate the capacitance from energy of the cell or using constant power discharge.

Our response: Thanks for the reviewer's valuable comment and kind suggestion. We agree with the reviewer's comment that the energetic efficiency should be provided in this system. Accordingly, we supplemented the energetic efficiency of our EHGC in the acidic electrolyte, which delivers a value of 71% at a current density of 4 A g⁻¹. However, we would like to point out that Coulombic efficiency in our EHGC systems is also very important to evaluate the reversibility of the cell reactions, which is widely reported in the hybrid capacitor systems (Adv. Energy Mater. 2020, 10, 2001705; Adv. Energy Mater. 2018, 8, 1703043).

Our revision: According to the reviewer's comment, we made changes to the revised manuscript as shown in the following. The detailed revision is shown as follows: "*The EHGC in the acidic electrolyte delivers an energetic efficiency of 71% at a current density of 4 A g⁻¹*" Please see the highlighted part in red color on page 7 in the revised manuscript.

Comment 8: In order to show that the concept works as energy storage system, self-discharge and leakage currents should be presented.

Our response: We appreciate the reviewer's good comment. We agree with the reviewer's comment that self-discharge and leakage current are also very vital for our energy storage system. Accordingly, we tested the self-discharge performance of our EHGC at 1 A g^{-1} . The EHGC in the acidic electrolyte is capable of retaining $\sim 79\%$ of the initial capacity with the leakage current of 10 mA g^{-1} after 2 h of self-discharge (Fig. R4a, b). The EHGC in the neutral electrolyte is capable of retaining $\sim 82\%$ of the initial capacity with the leakage current of 2.2 mA g^{-1} after 5 h of self-discharge (Fig. R4c, d). The EHGC in the alkaline electrolyte is capable of retaining $\sim 83\%$ of the initial capacity with the leakage current of 1.2 mA g^{-1} after 10 h of self-discharge (Fig. R4e, f). The leakage current (I_l , mA g^{-1}) of the EHGCs was calculated from the charge/discharge curves after self-discharge based on the equation of $I_l = (Q_c - Q_d)/t_s$, where Q_c is charge capacity, Q_d is discharge capacity, and t_s is self-discharge time.

Figure R2. The self-discharge behavior of the EHGC. (a) The voltage-time and (b) voltage-capacity curves at a current density of 1 A g^{-1} in the $9 \text{ M H}_3\text{PO}_4$ electrolyte. (c) The voltage-time and (d) voltage-capacity curves at a current density of 1 A g^{-1} in the 1 M PBES electrolyte. (e) The voltage-time and (f) voltage-capacity curves at a current density of 1 A g^{-1} in the 2 M KOH electrolyte.

Comment 9: THE CAPACITANCE OF CAPACITORS IS CONSTANT REGARDLESS VOLTAGE. How to understand the profiles of voltage vs. capacitance???? Authors should look at CV profiles and then re-consider their graphs.

Our response: Thanks a lot for pointing out this part by the reviewer. We have come to realize that the capacitance of capacitors does not change along with the voltage. Accordingly, we have changed all figures about voltage-capacitance curves into the graphs of voltage-time curves. Figure 2-5 have been updated in the revised manuscript.

Fig. 2. Electrochemical performance of the EHGCs in acidic electrolytes. a CV curves of EHGC and EDLC at the scan rate of 10 mV s^{-1} . **b** Charge/discharge curves of EHGC and

EDLC at a current density of 1 A g^{-1} . **c** Charge/discharge curves at a voltage range of 0-1.2 V at different current densities. **d** Capacitance (F g^{-1}) and specific capacity (mAh g^{-1}) as a function of the current density of EHGC and EDLC. **e** Cycling performance at a current density of 20 A g^{-1} . **f** Cycling performance under different atmosphere at a current density of 1 A g^{-1} . The comparison of **g** rate capacitance and **h** cycling retention ratio between different supercapacitors and the EHGC.

Fig. 3. Electrochemical performance of the EHGCs in neutral electrolytes. a CV curves at different scan rates. **b** Charge/discharge curves in different voltage ranges at a current density of 1 A g^{-1} . **c** Charge/discharge curves in a voltage range of 0-1.3 V at different current densities. **d** CV curves of EHGC and EDLC at the scan rate of 10 mV s^{-1} . **e** Charge/discharge curves of EHGC and EDLC at a current density of 1 A g^{-1} . **f** Capacitance and capacity as a function of current density for EHGC and EDLC. **g** Cycling performance at a current density of 10 A g^{-1} . Inset shows GCD curves in the 1st, 10,000th, and 100,000th cycles.

Fig. 4. Electrochemical performance of the EHGCs in alkaline electrolytes. a CV curves of EHGC and EDLC at a scan rate of 10 mV s⁻¹. **b** Charge/discharge curves of EHGC and EDLC at a current density of 1 A g⁻¹. **c** Charge/discharge curves in a voltage range of 0-1.1 V at different current densities. **d** Capacitance and specific capacity as a function of current density for the EHGC and EDLC. **e** Cycling performance at a current density of 10 A g⁻¹.

Fig. 5. Electrochemical performance of the EHGCS in all climates. **a** Charge/discharge curves with a current density of 1 A g^{-1} at various operating temperatures from $-50 \text{ }^{\circ}\text{C}$ to $40 \text{ }^{\circ}\text{C}$. Charge/discharge curves with various current densities at **b** $60 \text{ }^{\circ}\text{C}$, **c** $-60 \text{ }^{\circ}\text{C}$, and **d** $-70 \text{ }^{\circ}\text{C}$. **e** Capacitance and specific capacity as a function of the temperature at various current densities. **f** Cycling performance at 4 A g^{-1} at $-20 \text{ }^{\circ}\text{C}$. The EHGCS were tested in the acidic electrolyte of $9 \text{ M H}_3\text{PO}_4$.

Comment 10: What is the benefit of the system if the voltage is limited to 1.2 V ?

Our response: Thanks a lot for the reviewer's insightful comment. Although the voltage of our EHGCS is limited to 1.2 V in aqueous electrolytes, they show some unique advantages as listed in the followings. 1) The EHGCS are able to achieve higher capacities for the AC positive electrode in comparison with the conventional EDLCs. Due to the much higher capacity of the negative hydrogen gas electrode over the conventional carbon electrode, the EHGCS can deliver much higher cell capacity and energy density than the EDLCs. For example, our constructed EHGCS exhibits a cell capacitance of 295 F g^{-1} at 1 A g^{-1} , which is much higher than that of 109 F g^{-1} for the EDLC. The energy density of 45 Wh kg^{-1} of our

EHGC is about 4.5 times higher than the EDLC with the energy density of 10 Wh kg^{-1} . 2) Most of the reported hybrid capacitors displayed limited rate and cycle life (e.g. zinc-ion capacitor) due to the poor electrochemical property of the negative electrode. However, our EHGCs deliver an ultralong cycling stability for over 100000 cycles and fast charge/discharge rates up to 30 A g^{-1} . 3) Our EHGCs can be operated well in the full pH range from 0 to 14 and a wide temperature range from $-70 \text{ }^\circ\text{C}$ to $60 \text{ }^\circ\text{C}$.

Comment 11: What are the capacitance values of single electrodes? And how these electrodes were balanced? Having 50 Wh/kg at 1.2 V it seems that the capacitance reaches 1000 F/g ($\sim 330 \text{ mAh/g}$).

Our response: Thanks for the reviewer's valuable comment. To test the capacitance value of the single AC electrode, we have conducted a three-electrode measurement by using AC as the working electrode, graphite rod as the counter electrode, and Ag/AgCl as the reference electrode in the $9 \text{ M H}_3\text{PO}_4$ electrolyte. As shown in Fig. R3, the capacitance of the single AC electrode at a current density of 1 A g^{-1} is about 304 F g^{-1} , which is close to the cell capacitance value of the EHGC (295 F g^{-1}) at the same current density (Fig. 3b). This indicates that the introduction of hydrogen gas electrode does not affect the adsorption/desorption ability of the AC positive electrode. Hydrogen gas electrode with a low mass loading (0.2 mg cm^{-2}) of catalyst can provide much larger capacity (theoretical capacity of 2976 mAh g^{-1} based on the mass of H_2O and the reaction of $2\text{H}_2\text{O} + 2\text{e}^- \leftrightarrow \text{H}_2 + 2\text{OH}^-$) than the AC positive electrode. Therefore, in our EHGC system, the capacity of the negative electrode does not need to balance with the positive electrode. In other words, the capacity of our EHGC is mostly determined by the positive electrode. The specific capacity (Q , in the unit of mAh g^{-1}) and specific capacitance (C , in the unit of F g^{-1}) of the EHGCs and EDLCs were calculated from the discharge curves based on the following equations:

$$Q = It/m$$

$$C = 3.6 * Q/U$$

where I is the current, t is the discharge time, U is the testing voltage range, and m is the mass of active material from the positive electrode.

The energy density (E) and power density (P) of the EHGCs and EDLCs were calculated based on the following equations:

$$E = \int_{t_1}^{t_2} IV/mdt$$

where Δt is the discharge time, I is the current, V is the voltage after Ohmic drop, and m is the mass of active materials from the positive and negative electrodes. Therefore, in the ideal situation, the energy density (E) can be approximately calculated by the equation of $E = 0.5 * QU$. Therefore, having 50 Wh/kg at 1.2 V the capacitance can reach 250 F/g (~83 mAh/g).

Fig. R3. Charge/discharge curve of single AC positive electrode at a current density of 1 A g⁻¹ in the 9 M H₃PO₄ electrolyte.

Our revision: According to the reviewer’s comment, we made changes to the revised manuscript as shown in the following. In the revised manuscript, we added the discussion as follows: “*The specific capacity (Q , in the unit of mAh g⁻¹) and specific capacitance (C , in the unit of F g⁻¹) of the EHGCs and EDLCs were calculated from the discharge curves based on the following equations²²:*”

$$Q = It/m$$

$$C = 3.6 * Q/U$$

where I is the current, t is the discharge time, U is the testing voltage range, and m is the mass of active material from the positive electrode.” Please see the highlighted part in red color on page 21 in the revised manuscript.

For all these reasons, I cannot recommend the paper for publishing in the present form, I recommend rejection and re-submission once seriously and correctly revised.

Our response: We are grateful to the reviewer’s comments and suggestions again, which are beneficial to improving the quality of our manuscript. We have tried our best to conduct additional experiments and make correct expression in the revised manuscript and supporting

information. All changes have been highlighted in red color in the revised manuscript. We hope that the revised manuscript is suitable for publication in *Nature Communications*.

REVIEWERS' COMMENTS

Reviewer #1 (Remarks to the Author):

The authors have answered to my comments.

Reviewer #2 (Remarks to the Author):

The revised version of the manuscript correctly addressed the remarks and doubts raised by the Referee. Furthermore, Authors have introduced several modifications that made the work more clear, convincing and reproducible. Thus, I recommend the paper for publishing.

Response to Reviewers' Comments

Reviewer #1: The authors have answered to my comments.

Our response: We are grateful to the reviewer's positive comment.

Reviewer #2: The revised version of the manuscript correctly addressed the remarks and doubts raised by the Referee. Furthermore, Authors have introduced several modifications that made the work more clear, convincing and reproducible. Thus, I recommend the paper for publishing.

Our response: We are very grateful to the reviewer's highly positive comment.